# Identifying Learning Rules From Neural Network Observables

**Aran Nayebi[1,*], Sanjana Srivastava[2,*], Surya Ganguli[3,5], and Daniel L.K. Yamins[2,4,5]**

[1]Neurosciences Ph.D. Program, Stanford University
[2]Department of Computer Science, Stanford University
[3]Department of Applied Physics, Stanford University
[4]Department of Psychology, Stanford University
[5]Wu Tsai Neurosciences Institute, Stanford University
[*]Equal contribution. Correspondence: `anayebi@stanford.edu`

## Abstract

The brain modifies its synaptic strengths during learning in order to better adapt to its environment. However, the underlying plasticity rules that govern learning are unknown. Many proposals have been suggested, including Hebbian mechanisms, explicit error backpropagation, and a variety of alternatives. It is an open question as to what specific experimental measurements would need to be made to determine whether any given learning rule is operative in a real biological system. In this work, we take a "virtual experimental" approach to this problem. Simulating idealized neuroscience experiments with artificial neural networks, we generate a large-scale dataset of learning trajectories of aggregate statistics measured in a variety of neural network architectures, loss functions, learning rule hyperparameters, and parameter initializations. We then take a discriminative approach, training linear and simple non-linear classifiers to identify learning rules from features based on these observables. We show that different classes of learning rules can be separated solely on the basis of aggregate statistics of the weights, activations, or instantaneous layer-wise activity changes, and that these results generalize to limited access to the trajectory and held-out architectures and learning curricula. We identify the statistics of each observable that are most relevant for rule identification, finding that statistics from network activities across training are more robust to unit undersampling and measurement noise than those obtained from the synaptic strengths. Our results suggest that activation patterns, available from electrophysiological recordings of post-synaptic activities on the order of several hundred units, frequently measured at wider intervals over the course of learning, may provide a good basis on which to identify learning rules.

## 1 Introduction

One of the tenets of modern neuroscience is that the brain modifies its synaptic connections during learning to improve behavior [Hebb, 1949]. However, the underlying plasticity rules that govern the process by which signals from the environment are transduced into synaptic updates are unknown. Many proposals have been suggested, ranging from Hebbian-style mechanisms that seem biologically plausible but have not been shown to solve challenging real-world learning tasks [Bartunov et al., 2018]; to backpropagation [Rumelhart et al., 1986], which is effective from a learning perspective but has numerous biologically implausible elements [Grossberg, 1987, Crick, 1989]; to recent regularized circuit mechanisms that succeed at large-scale learning while remedying some of the implausibilities of backpropagation [Akrout et al., 2019, Kunin et al., 2020].

A major long-term goal of computational neuroscience will be to identify which of these routes is most supported by neuroscience data, or to convincingly identify experimental signatures that reject all of them and suggest new alternatives. However, along the route to this goal, it will be necessary to develop practically accessible experimental observables that can efficiently separate between hypothesized learning rules. In other words, what specific measurements – of activation patterns over time, or synaptic strengths, or paired-neuron input-output relations – would allow one to draw quantitatively tight estimates of whether the observations are more consistent with one or another specific learning rule? This in itself turns out to be a substantial problem, because it is difficult on purely theoretical grounds to identify which patterns of neural changes arise from given learning rules, without also knowing the overall network architecture and loss function target (if any) of the learning system.

In this work, we take a "virtual experimental" approach to this question, with the goal of answering whether it is even possible to generically identify which learning rule is operative in a system, across a wide range of possible learning rule types, system architectures, and loss targets; and if it is possible, which types of neural observables are most important in making such identifications. We simulate idealized neuroscience experiments with artificial neural networks trained using different learning rules, across a variety of architectures, tasks, and associated hyperparameters. We first demonstrate that the learning rules we consider can be reliably separated *without* knowledge of the architecture or loss function, solely on the basis of the trajectories of aggregate statistics of the weights, activations, or instantaneous changes of post-synaptic activity relative to pre-synaptic input, generalizing as well to unseen architectures and training curricula. We then inject realism into how these measurements are collected in several ways, both by allowing access to limited portions of the learning trajectory, as well as subsampling units with added measurement noise. Overall, we find that measurements temporally spaced further apart are more robust to trajectory undersampling. We find that aggregated statistics from recorded activities across training are most robust to unit undersampling and measurement noise, whereas measured weights (synaptic strengths) provide reliable separability as long as there is very little measurement noise but can otherwise be relatively susceptible to comparably small amounts of noise.

## 2 Related Work

Lim et al. [2015] infer a Hebbian-style plasticity rule from IT firing rates recorded in macaques to novel and familiar stimuli during a passive and an active viewing task. By assuming that the plasticity rule is a separable function of pre-synaptic and post-synaptic rates acting on a non-linear recurrent rate model, their result demonstrates that one can infer the *hyperparameters* of a learning rule from post-synaptic activities alone, given a specific architecture (e.g. recurrent excitatory to excitatory connections) and *single class* of learning rule (e.g. Hebbian).

Since Grossberg [1987] introduced the *weight transport problem*, namely that backpropagation requires exact transposes to propagate errors through the network, many credit assignment strategies have proposed circumventing the problem by introducing a distinct set of feedback weights to propagate the error backwards. Broadly speaking, these proposals fall into two groups: those that encourage symmetry between the forward and backward weights [Lillicrap et al., 2016, Nøkland, 2016, Liao et al., 2016, Xiao et al., 2019, Moskovitz et al., 2018, Akrout et al., 2019, Kunin et al., 2020], and those that encourage preservation of information between neighboring network layers [Bengio, 2014, Lee et al., 2015, Kunin et al., 2020].

However, Kunin et al. [2020] show that temporally-averaged post-synaptic activities from IT cortex (in adult macaques during passive viewing of images) are not sufficient to separate many different *classes* of learning rules, even for a fixed architecture. This suggests that neural data most useful for distinguishing between classes of learning rules in an *in vivo* neural circuit will likely require more temporally-precise measurements during learning, but does not prescribe what quantities should be measured.

In artificial networks, every unit can be measured precisely and the ground truth learning rule is known, which may provide insight as to what experimentally measurable observables may be most useful for inferring the underlying learning rule. In the case of single-layer linear perceptrons with Gaussian inputs, the error curves for a mean-squared error loss can be calculated exactly from the unit-averaged weight updates [Baldi and Hornik, 1989, Heskes and Kappen, 1991, Werfel et al.,

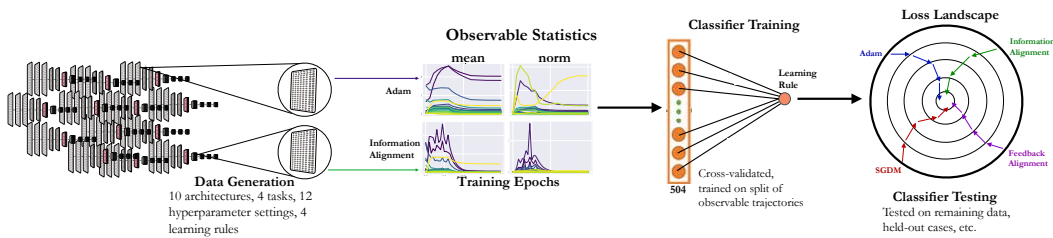

Figure 1: **Overall approach.** Observable statistics are generated from each layer of 1,056 neural networks, through the model training process for each learning rule. Qualitatively, trajectories show patterns that are common across the observable statistics and distinctive among learning rules. We take a quantitative approach whereby a classifier is cross-validated and trained on a subset of these trajectories and evaluated on the remaining data.

2004]. Of course, it is not certain whether this signal would be useful for multi-layered networks across multiple loss functions, architectures, and highly non-Gaussian inputs at scale.

In order to bear on future neuroscience experiments, it will be important to identify useful aggregate statistics (beyond just the mean) and how robust these observable statistics are when access to the full learning trajectory is no longer allowed, or if they are only computed from a subset of the units with some amount of measurement noise.

## 3 Approach

**Defining features.** The primary aim of this study is to determine the general separability of classes of learning rules. Fig. 1 indicates the general approach of doing so, and the first and second panels illustrate feature collection. In order to determine what needs to be measured to reliably separate classes of learning rules, we begin by defining representative features that can be drawn from the course of model training. For each layer in a model, we consider three measurements: **weights** of the layer, **activations** from the layer, and **layer-wise activity change** of a given layer's outputs relative to its inputs. We choose artificial neural network weights to analogize to synaptic strengths in the brain, activations to analogize to post-synaptic firing rates, and layer-wise activity changes to analogize to *paired* measurements that involve observing the change in post-synaptic activity with respect to changes induced by pre-synaptic input. The latter observable is motivated by experiments that result in LTP induction via tetanic stimulation [Wojtowicz and Atwood, 1985], in the limit of an infinitesimally small bin-width.

For each measure, we consider three functions applied to it: identity, absolute value, and square. Finally, for each function of the weights and activations, we consider seven statistics: Frobenius norm, mean, variance, skew, kurtosis, median, and third quartile. For the layer-wise activity change observable, we only use the mean statistic because instantaneous layer-wise activity change is operationalized as the gradient of outputs with respect to inputs (definable for *any* learning rule, regardless of whether or not it uses a gradient of an error signal to update the weights). The total derivative across neurons can be computed efficiently, but computing the derivative for every output neuron in a given layer is prohibitively computationally complex. This results in a total of 45 continuous valued observable statistics for each layer, though 24 observable statistics (listed in Fig. 3) are ultimately used for training the classifiers, since we remove any statistic that has a divergent value during the course of model training. We also use a ternary indicator of **layer position** in the model hierarchy: "early", "middle", or "deep" (represented as a one-hot categorical variable).

**Constructing a varied dataset.** We use the elements of a trained model other than learning rule as factors of variation. Specifically, we vary architectures, tasks, and learning hyperparameters. The tasks and architectures we consider are those that have been shown to produce good representations to (mostly primate) sensory neural and behavioral data [Yamins et al., 2014, Kell et al., 2018, Nayebi et al., 2018, Schrimpf et al., 2018, Cadena et al., 2019, Feather et al., 2019]. Specifically, we consider the tasks of **supervised 1000-way ImageNet categorization** [Deng et al., 2009], *self-supervised* **ImageNet** in the form of the recent competitively performing "SimCLR" contrastive loss

and associated data augmentations [Chen et al., 2020], **supervised 794-way Word-Speaker-Noise (WSN) categorization** [Feather et al., 2019], and **supervised ten-way CIFAR-10 categorization** [Krizhevsky, 2010]. We consider six architectures on ImageNet, SimCLR, and WSN, namely, ResNet-18, ResNet-18v2, ResNet-34, and ResNet-34v2 [He et al., 2016], as well as Alexnet [Krizhevsky et al., 2012] both without and with local response normalization (denoted as Alexnet-LRN). On CIFAR-10, we consider four shallower networks consisting of either 4 layers with local response normalization, due to Krizhevsky [Krizhevsky, 2012], or a 5 layer variant (denoted as KNet4-LRN and KNet5-LRN, respectively), as well as without it (denoted as KNet4 and KNet5). The latter consideration of these networks on CIFAR-10 is perhaps biologically interpretable as expanding scope to shallower *non-primate* (e.g. mouse) visual systems [Harris et al., 2019].

The dependent variable is the learning rule (across hyperparameters), and we consider four: **stochastic gradient descent with momentum (SGDM)** [Sutskever et al., 2013], **Adam** [Kingma and Ba, 2014], **feedback alignment (FA)** [Lillicrap et al., 2016], and **information alignment (IA)** [Kunin et al., 2020]. Learning hyperparameters for each model under a given learning rule category are the Cartesian product of three settings of batch size (128, 256, and 512), two settings of the random seed for architecture initialization (referred to as "model seed"), and two settings of the random seed for dataset order (referred to as "dataset randomization seed"). The base learning rate is allowed to vary depending on what is optimal for a given architecture on a particular dataset and is rescaled accordingly for each batch size setting. All model training details can be found in Appendix A.

We select these learning rules as they span the space of commonly used variants of backpropagation (SGDM and Adam), as well as potentially more biologically-plausible "local" learning rules that efficiently train networks at scale to varying degrees of performance (FA and IA) and avoid weight transport. We do not explicitly train models with more classic learning rules such as pure Hebbian learning or weight/node perturbation [Widrow and Lehr, 1990, Jabri and Flower, 1992], since for the multi-layer non-linear networks that we consider here, those learning rules are either highly inefficient in terms of convergence (e.g. weight/node perturbation [Werfel et al., 2004]) or they will readily fail to train due to a lack of stabilizing mechanism (e.g. pure Hebbian learning). However, the local learning rules rules we do consider (FA and IA) incorporate salient aspects of these algorithms in their implementation – IA incorporates Hebbian and other stabilizing mechanisms in its updates to the feedback weights and FA employs a pseudogradient composed of random weights reminiscent of weight/node perturbation.

To gain insight into the types of observables that separate learning rules in general, we use statistics (averaged across the validation set) generated from networks trained across 1,056 experiments as a feature set on which to train simple classifiers, such as a linear classifier (SVM) and non-linear classifiers (Random Forest and a Conv1D MLP) on various train/test splits of the data. This is illustrated in the third and fourth panels of Fig. 1, with the feature and parameter selection procedure detailed in Appendices B.1 and B.2.

## 4 Results on Full Dataset

In this section, we examine learning rule separability when we have access to the entire training trajectories of the models, as well as noiseless access to *all* units. This will allow us to assess whether the learning rules can be separated independent of loss function and architecture, what types of observables and their statistics are most salient for separability, and how robust these conclusions are to certain classes of input types to explore strong generalization.

**General separability problem is tractable.** For each observable measure, we train the classifier on the concatenation of the trajectory of statistics identified in §3, generated from each model layer. Already by eye (Fig. 1), one can pick up distinctive differences across the learning rules for each of the training trajectories of these metrics. Of course, this is not systematic enough to clearly judge one set of observables versus another, but provides some initial assurance that these metrics seem to capture some inherent differences in learning dynamics across rules.

As can be seen from Fig. 2(a), across all classes of observables, the Random Forest attains the highest test accuracy, and all observable measures perform similarly under this classifier, even when distinguishing between any given pair of learning rules (Fig. S1). The Conv1D MLP never outperforms the Random Forest, and only slightly outperforms the SVM in most cases, despite having the capability of learning additional non-linear features beyond the input feature set.

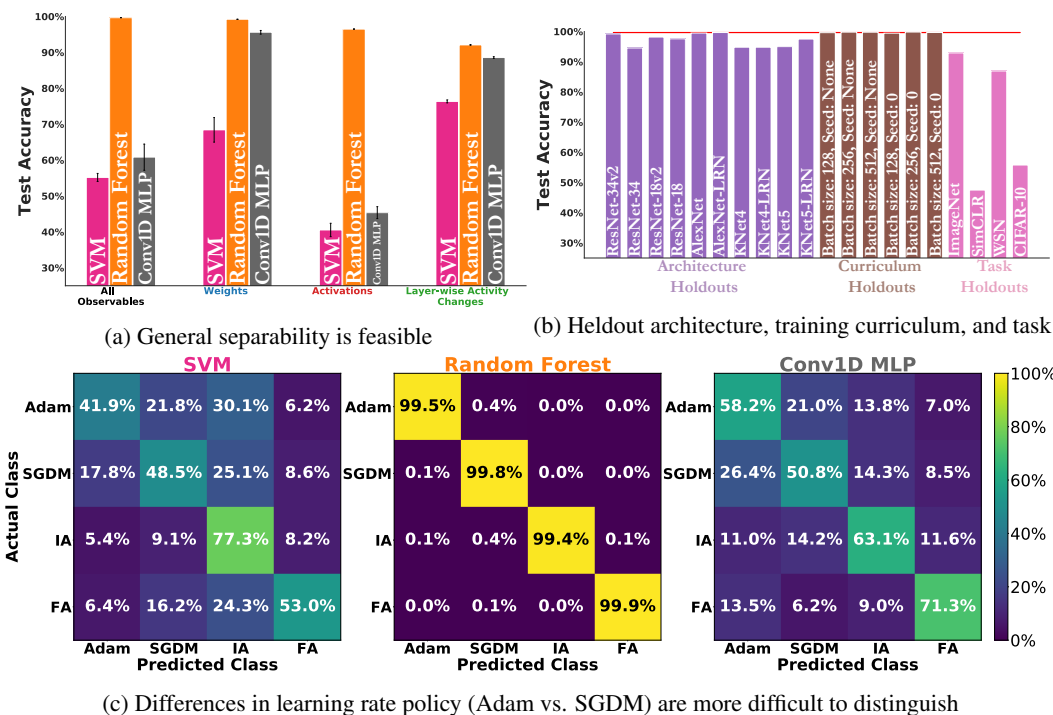

(a) General separability is feasible

(b) Heldout architecture, training curriculum, and task

(c) Differences in learning rate policy (Adam vs. SGDM) are more difficult to distinguish

Figure 2: **Quantifying successful separation. (a)** shows the test accuracy of each classifier, with mean and s.e.m. across ten category-balanced 75%/25% train/test splits, using the observable measures in §3. **(b)** Let the red line indicate the mean and s.e.m. test accuracy of the Random Forest trained on all observable measures in (a). The left-most violet bars indicate Random Forest performance when holding out *all* examples from each of the ten architectures we consider, for each observable measure. The middle brown bars indicate Random Forest performance when holding out *all* examples from each of the six combinations of batch size and dataset randomization seed pair. The right-most pink bars indicate Random Forest performance when holding out *all* examples from each of the four tasks. **(c)** Confusion matrices on the test set (1,296 examples per class), averaged across the ten category-balanced 75%/25% train/test splits, for each of the three classifiers when trained on all observable measures in (a). Chance performance in these settings is 25% test accuracy.

In fact, from the confusion matrices in Fig. 2(c), the Random Forest hardly mistakes one learning rule from any of the others. However, when the classifiers make mistakes, they tend to confuse Adam vs. SGDM more so than IA vs. FA, suggesting that they are able to pick up more on differences, reflected in the observable statistics, due to high-dimensional direction of the gradient tensor than the magnitude of the gradient tensor (the latter directly tied to learning rate policy).

In terms of linear separability, the SVM can attain its highest test accuracy using either the weight or layer-wise activity change observables. Interestingly, the layer-wise activity change observable has the least number of statistics (three) associated with it compared to the weight and activation observables. The class of learning rule, however, is least able to be *linearly* determined from the activation observables. The latter observation holds as well when distinguishing between any given pair of learning rules (Fig. S1).

To further gain insight into drivers of learning rule discriminability, we consider two additional sets of controls, namely that of task performance and the scale of observable statistics. In Fig. S2(a), we find that where defined, task performance is a confounded indicator of separability. SGDM and IA are easily separated by classifiers trained on any observable measure, despite yielding similar top-1 validation accuracy on ImageNet across ResNet architectures and hyperparameters.

In Fig. S2(b), we train classifiers on observable statistics averaged over the whole learning trajectory. We find that averaging the trajectory hurts performance, relative to having the full trajectory, for the (linear) SVM classifier across all classes of observables. This is also the case for the Random Forest, but far less for the weight observable statistics, suggesting this type of non-linear classifier

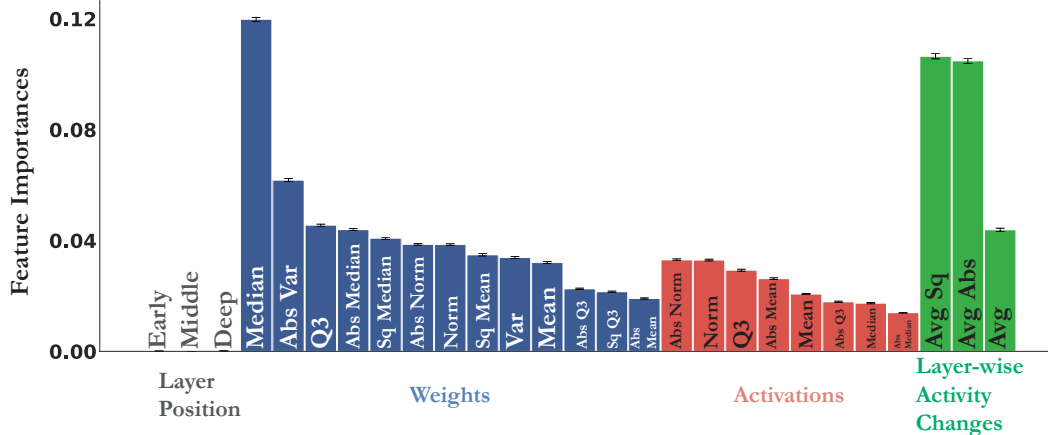

Figure 3: **Relative importances of observable statistics.** We show the Gini impurity feature importance (summed across the learning trajectory) of each observable statistic for the Random Forest classifier trained on all observable measures in Fig. 2(a). The colors indicate observable measures, demonstrating the prevalence and importance of observable statistics from a given measure. Mean and s.e.m. across trees in the Random Forest and ten category-balanced 75%/25% train/test splits. "Sq" and "Abs" indicate squaring or taking the absolute value across all units prior to computing the statistic, respectively. "Q3" is the abbreviation for the third quartile statistic.

has enough information from the trajectory-averaged weight statistics during the course of learning to decode the identity of the learning rule. Overall, this result suggests that the scale of the observable statistics is not sufficient in all cases to distinguish learning rules, though it certainly contributes in part to it.

**Generalization to entire held-out classes of input types.** The results so far were obtained across ten category-balanced splits of examples with varying tasks, model architectures, and training curricula. However, it is not clear from the above whether our approach will generalize to *unseen* architectures and training curricula. In particular, if a new "animal" were added to our study, we ideally do not want to be in a position to have to completely retrain the classifier or have the conclusions change drastically, nor do we want the conclusions to be highly sensitive to training curricula (the latter analogized in this case to batch size and dataset randomization seed).

As seen in Fig. 2(b), the performance of the Random Forest for the held out conditions of architecture and training curriculum is not much worse than its corresponding test set performance in Fig. 2(a), indicated by the red line. This relative robustness across architectures and training curricula holds even if you train on either the weight or activation observable measures individually (Fig. S3(a,b)), but not as consistently with the layer-wise activity changes (Fig. S3(a)) or with a linear classifier (Fig. S4(a,b)).

We also tested held-out task to quantify differences in learning dynamics between different tasks, although it is not necessarily reflective of a real experimental condition where task is typically fixed. In particular, generalizing to deep models trained on ImageNet or WSN works the best, despite them being different sensory modalities. However, generalizing from the supervised cross-entropy loss to the self-supervised SimCLR loss performs the lowest, as well as generalizing from deep architectures on ImageNet, SimCLR, and WSN to shallow architectures on CIFAR-10 (in the latter case, both task and architecture have changed from the classifier's train to test set). This quantifies potentially fundamental differences in learning dynamics between supervised vs. self-supervised tasks and between deep networks on large-scale tasks vs. shallow networks on small-scale tasks.

Taken together, Fig. 2 suggests that a simple non-linear classifier (Random Forest) can attain consistently high generalization performance when trained across a variety of tasks with either the weight or activation observable measures *alone*.

**Aggregate statistics are not equally important.** We find from the Random Forest Gini impurity feature importances for each measure's individual statistic that not all the aggregate statistics within a given observable measure appear to be equally important for separating learning rules, as displayed

in Fig. 3. For the weights, the median is given the most importance, even in an absolute sense across all other observable measures. For the activations, the norm statistics are given the most importance for that measure. For the averaged (across output units) layer-wise activity changes, the magnitude of this statistic either in terms of absolute value or square are assigned similar importances. On the other hand, the first three features comprising the ternary categorical "layer position" are assigned the lowest importance values.

Of course, we do not want to over-interpret these importance values given that Gini impurity has a tendency to bias continuous or high cardinality features. The more computationally expensive permutation feature importance could be less susceptible to such bias. However, these results are suggestive that for a given observable measure, there may only be a small subset of statistics most useful for learning rule separability, indicating that in a real experiment, it might not be necessary to compute very many aggregate statistics from recorded units so long as a couple of the appropriate ones for that measure (e.g. those identified from classifiers) are used. While no theory of neural networks yet allows us to derive optimal statistics mathematically (motivating our empirical approach), ideally in the future we can combine better theory with our method to sharpen feature design.

## 5 Access to Only Portions of the Learning Trajectory

The results in §4 were obtained with access to the entire learning trajectory of each model. Often however, an experimentalist collects data throughout learning at regularly spaced intervals [Holtmaat et al., 2005]. We capture this variability by randomly sampling a fixed number of samples at a fixed spacing ("subsample period") from each trajectory, as is done in Fig. 4(a). See Appendix B.3.1 for additional details on the procedure.

We find across observable measures that robustness to undersampling of the trajectory is largely dependent on the subsample period length (number of epochs between consecutive samples). As the subsample period length increases (middle and right-most columns of Fig. 4(a)), the Random Forest classification performance[1] increases compared to the *same* number of samples for a smaller period (left-most column).

To further demonstrate the importance of subsampling widely *across* the learning trajectory, we train solely on a consecutive third of the trajectory ("early", "middle", or "late") and separately test on each of the remaining consecutive thirds in Fig. 4(b). This causes a reduction in generalization performance relative to training and testing on the *same* consecutive portion (though there is some correspondence between the middle and late portions). For a fixed observable measure, even training and testing on the same consecutive portion of the learning trajectory does not give consistent performance across portions (Fig. S6). The use of a non-linear classifier is especially crucial in this setting, as the (linear) SVM often attains chance test accuracy or close to it (Fig. S7).

Taken together, these results suggest that data that consists of measurements collected temporally further apart across the learning trajectory is more robust to undersampling than data collected closer together in training time. Furthermore, across *individual* observable measures, the *weights* are overall the most robust to undersampling of the trajectory, but with enough frequency of samples we can achieve comparable performance with the activations.

## 6 Unit Undersampling and Measurement Noise Robustness of Observables

The aggregate statistics computed from the observable measures have thus far operated under the idealistic assumption of noiseless access to every unit in the model. However, in most datasets, there is a significant amount of unit undersampling as well as non-zero measurement noise. How do these two factors affect learning rule identification, and in particular, how noise and subsample-robust are particular observable measures? Addressing this question would provide insight into the types of experimental paradigms that may be most useful for identifying learning rules, and predict how certain experimental tools may fall short for given observables.

To assess this question, we consider an anatomically reasonable feedforward model of primate visual cortex, ResNet-18, trained on both supervised ImageNet and the self-supervised SimCLR

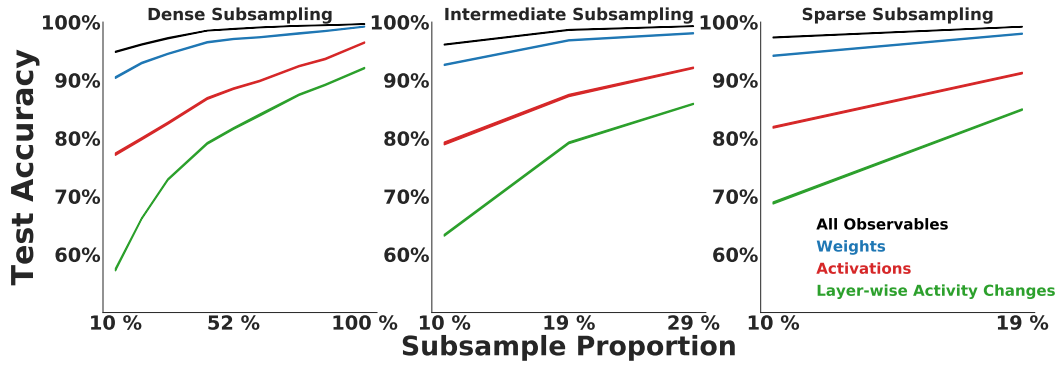

(a) Sparse subsampling *across* the learning trajectory is robust to trajectory undersampling.

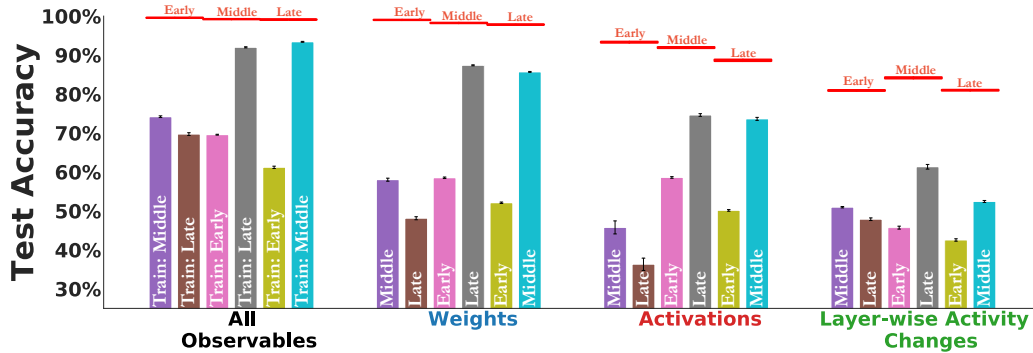

(b) Training solely on consecutive portions of the learning trajectory is *not* robust to trajectory undersampling.

Figure 4: **Trajectory subsampling. (a)** Sparse subsampling, where the epochs between nearby trajectory samples are the furthest apart (right panel; 25 epochs apart), requires far fewer samples to achieve the same performance for the Random Forest as the full trajectory (left panel; 5 epochs apart). "Subsample Proportion $Y$%" refers to the number of samples chosen for the trajectory subsample relative to that of the full trajectory (21 total samples). **(b)** We train the Random Forest on 75% of examples with access to only one consecutive third of the full trajectory (7 consecutive samples each out of 21 total samples), testing on the remaining 25% of examples from one of the other two consecutive thirds. Red lines denote Random Forest performance when tested on 25% of examples from the *same* portion of the trajectory as in classifier training for each observable measure, reported in the bottom right row of Fig. S6. Mean and s.e.m. in all cases are across ten category-balanced train/test splits. Chance performance in these settings is 25% test accuracy.

tasks (known to give neurally-plausible representations [Nayebi et al., 2018, Schrimpf et al., 2018, Zhuang et al., 2020]). We train the network with FA and IA, for which there have been suggested biologically-motivated mechanisms [Guerguiev et al., 2017, Lansdell and Kording, 2019, Kunin et al., 2020].

We model measurement noise as a standard additive white Gaussian noise (AWGN) process applied independently per unit with five values of standard deviation (including the case of no measurement noise). We subsample units from a range that is currently used by electrophysiological techniques[2] up to the case of measuring all of the units. See Appendix B.4 for additional details on the procedure.

From Fig. 5, we find that both the weight and layer-wise activity change observables are the most susceptible to unit undersampling and noise in measurements. The activations observable is the most robust across all noise and unit subsampling levels (regardless of classifier, see Fig. S8 for the SVM).

This data suggests that if one collects experimental data by imaging synaptic strengths, it is still crucial that the optical imaging readout not be very noisy, since even with the amount of units

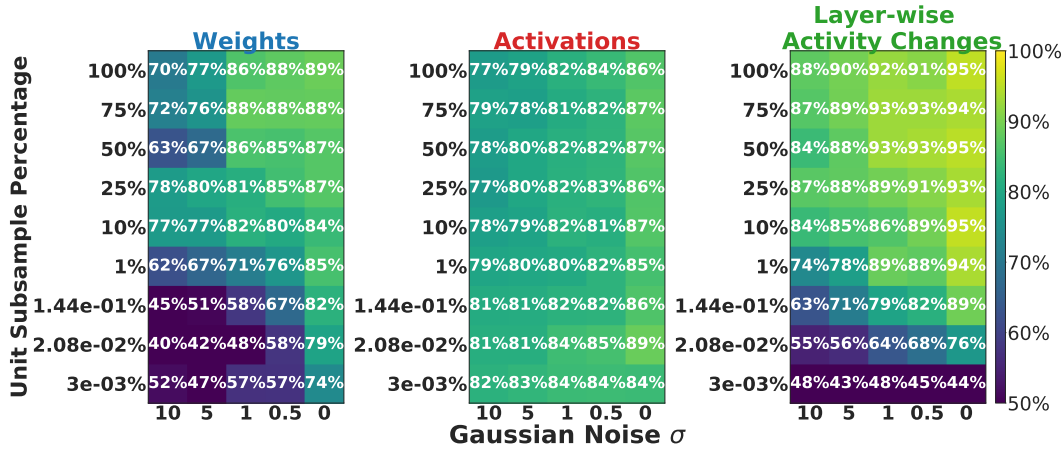

Figure 5: **Activations are the most robust to measurement noise and unit undersampling.** For each observable measure, the Random Forest test accuracy separating FA vs. IA on ResNet-18 across the ImageNet and SimCLR tasks, averaged across random seed and ten category-balanced 75%/25% train/test splits. The $x$-axis from the left to right of each heatmap corresponds to decreasing levels of noise, and the $y$-axis from bottom to top corresponds to increasing levels of sampled units. Top right corner of each heatmap corresponds to the perfect information setting of §4. Chance performance in this setting is 50% test accuracy.

typically recorded currently (e.g. several hundred to several thousand synapses, $\sim 2.08 \times 10^{-2}\%$ or less), a noisy imaging strategy of synaptic strengths may be rendered ineffective. Instead, current electrophysiological techniques that measure the *activities* from hundreds of units ($\sim 3 \times 10^{-3}\%$) could form a good set of neural data to separate learning rules. Recording more units with these techniques can improve learning rule separability from the activities, but it does not seem necessary, at least in this setting, to record a *majority* of units to perform this separation effectively.

## 7 Discussion

In this work, we undertake a "virtual experimental" approach to the problem of identifying measurable observables that may be most salient for separating hypotheses about synaptic updates in the brain. We have made this dataset publicly available[3], enabling others to analyze properties of learning rules without needing to train neural networks.

Through training 1,056 neural networks at scale on both supervised and self-supervised tasks, we find that across architectures and loss functions, it is possible to reliably identify what learning rule is operating in the system only on the basis of aggregate statistics of the weights, activations, or layer-wise activity changes. We find that with a simple non-linear classifier (Random Forest), each observable measure forms a relatively comparable feature set for predicting the learning rule category.

We next introduced experimental realism into how these observable statistics are measured, either by limiting access to the learning trajectory or by subsampling units with added noise. We find that measurements temporally spaced further apart are more robust to trajectory undersampling across all observable measures, especially compared to those taken at consecutive portions of the trajectory. Moreover, aggregate statistics across units of the network's activation patterns are most robust to unit undersampling and measurement noise, unlike those obtained from the synaptic strengths alone.

Taken together, our findings suggest that *in vivo* electrophysiological recordings of post-synaptic activities from a neural circuit on the order of several hundred units, frequently measured at wider intervals during the course of learning, may provide a good basis on which to identify learning rules. This approach provides a computational footing upon which future experimental data can be used to identify the underlying plasticity rules that govern learning in the brain.

## Broader Impact

Our work provides a basis as to what neuroscience experimental data should be collected in order to infer plasticity rules. Therefore, the experimental neuroscience community may benefit from this research. The data in this research is collected from neural networks trained on the ImageNet, Word-Speaker-Noise (WSN), and CIFAR-10 datasets. There is well-documented evidence of bias emerging from deep neural networks trained on field-standard large-scale image databases, with many ethical harms stemming from use of the biased algorithms to make important decisions on social policy or limit civil liberties. Rather than using network classifications in a real-world context, our study uses observable measurements from within the network mechanism as simulated brain data. This data is not interpretable to humans and is ultimately used to train and test a classifier to separate learning rules, which are a mathematical formulation and agnostic to any community. This study therefore does not leverage any bias in the data. However, identifying neural network observables that distinguish learning rules presents obvious privacy concerns which could be exploited, for example, to generate adversarial attacks or even to recover training data. It is possible that the observable statistics reflect the workings of a biased "*in silico* brain", in which case the classifier used to separate learning rules may be biased in that it has only been trained and tested on networks whose training data may be biased differently than the training data animals typically receive. This is a problem not just for this study, but generally comes down to available network-training tasks, especially those of the scale needed to provide neurally-plausible representations. Tasks are the most computationally complex factor of variation to add values to, so diversifying the set of tasks is an ongoing effort, and we hope to use balanced, ethically verified large-scale datasets as they become available. Furthermore, given the limited number of subjects in neuroscience experiments, this type of bias is not new in our experiment; the scalability of our approach may provide a way to have more representative data. The only consequence of failure is that the experiment which is performed on the basis of the conclusions we draw may not be as decisive in determining the learning rule.

## Acknowledgments and Disclosure of Funding

We thank Katherine L. Hermann, Eshed Margalit, and Jinyao Yan for helpful discussions. We thank the anonymous reviewers for their comments and suggestions. This work was funded in part by the IBM-Watson AI Lab. S.S. is supported by the National Science Foundation Graduate Research Fellowship Program (NSF GRFP). S.G. is supported by the James S. McDonnell Foundation, Simons Foundation, and National Science Foundation CAREER Award. D.L.K.Y is supported by the James S. McDonnell Foundation (Understanding Human Cognition Award Grant No. 220020469), the Simons Foundation (Collaboration on the Global Brain Grant No. 543061), the Sloan Foundation (Fellowship FG-2018-10963), the National Science Foundation (RI 1703161 and CAREER Award 1844724), the DARPA Machine Common Sense program, and hardware donation from the NVIDIA Corporation. We thank the Google TensorFlow Research Cloud (TFRC) team for generously providing TPU hardware resources for this project.

## Footnotes

[1]The advantage of larger subsample period lengths holds for a linear classifier such as the SVM (Fig. S5), but as expected, absolute performance is lower.

[2]Based on estimates of the total neuronal density in visual cortex being on the order of hundreds of millions [DiCarlo et al., 2012], compared to the number of units typically recorded on the order of several hundred [Majaj et al., 2015, Kar et al., 2019].

[3]`https://github.com/neuroailab/lr-identify`

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
