[Supplementary Material]

# A Model Training Details

We use TensorFlow version 1.13.1 to conduct all model training experiments. All model function code can be found here: `https://github.com/neuroailab/lr-identify/tree/main/tensorflow/Models`. The names of the model layers from which we generate observable statistics can be found here: `https://github.com/neuroailab/lr-identify/blob/main/tensorflow/Models/model_layer_names.py`.

The v2 variants of ResNets use the pre-activation of the weight layers rather than the post-activation used in the original versions [He et al., 2016]. Furthermore, the v2 variants of ResNets apply batch normalization [Ioffe and Szegedy, 2015] and ReLU to the input *prior* to the convolution, whereas the original variants apply these operations after the convolution. We use the TF Slim architectures for these two variants provided here: `https://github.com/tensorflow/models/tree/master/research/slim`.

Following Kunin et al. [2020], we replace tied weights in backpropagation with a regularization loss on untied forward and backward weights in order to train the alternative learning rules of FA [Lillicrap et al., 2016] and IA [Kunin et al., 2020] at scale. Each model layer consists of forward weights that parametrize the task objective function and the backward weights specify a descent direction, as implemented here: `https://github.com/neuroailab/neural-alignment`. The total network loss is defined as the sum of the task objective function $\mathcal{J}$ and the local alignment regularization $\mathcal{R}$. Setting $\mathcal{R} \equiv 0$ results in FA, and we use the same $\mathcal{R}$ and corresponding metaparameters for IA that were specified in Kunin et al. [2020]. SGDM refers to stochastic gradient descent with Nesterov momentum of 0.9 in all cases.

## A.1 ImageNet

This dataset [Deng et al., 2009] consists of 1,218,167 training images and 50,000 validation images from 1,000 total categories. The architectures trained on this task are AlexNet, AlexNet-LRN, ResNet-18, ResNet-18v2, ResNet-34, and ResNet-34v2. All models, except for the v2 ResNets, are trained on the standard ResNet preprocessing with $224 \times 224 \times 3$ sized ImageNet images. As used in the TF Slim repository linked above, the v2 ResNets are trained with Inception preprocessing, which uses larger $299 \times 299 \times 3$ sized images. The ResNets used an L2 regularization of $1 \times 10^{-4}$ and the AlexNets used an L2 regularization of $5 \times 10^{-4}$.

The base learning rate for SGDM and IA for the ResNet models is set to 0.125 for a batch size of 256, and linearly rescaled for the remaining batch sizes of 128 and 512. For AlexNet and AlexNet-LRN, this base learning rate is set an order of magnitude lower to 0.0125 since it does not employ batch normalization layers. For FA and Adam across all models, this base learning rate is 0.001 (in the case of FA, the Adam optimizer operates on the pseudogradient as is also done by Kunin et al. [2020]). Each base learning rate is linearly warmed up to its corresponding value for 6 epochs followed by 90% decay at 30, 60, and 80 epochs, training for 100 epochs total, as prescribed by Buchlovsky et al. [2019].

The AlexNet model function code can be found here: `https://github.com/neuroailab/lr-identify/blob/main/tensorflow/Models/alexnet.py`. The ResNet model function code for both variants can be found here: `https://github.com/neuroailab/lr-identify/blob/main/tensorflow/Models/resnet_model_google.py`.

## A.2 SimCLR

The architectures trained on this task are the same as in §A.1, namely, AlexNet, AlexNet-LRN, ResNet-18, ResNet-18v2, ResNet-34, and ResNet-34v2. Following Chen et al. [2020], for all models we use the same ImageNet preprocessing with $224 \times 224 \times 3$ sized images, L2 regularization of $1 \times 10^{-6}$, and two contrastive layers after dropping the final ImageNet categorization layer of the model, adapting their implementation provided here: `https://github.com/google-research/simclr/`. The only difference is that given that our batch sizes are small (between 128-512), we do not use the LARS optimizer [You et al., 2017] nor do we aggregate batch examples and batch norm statistics across TPU shards, in order to keep the learning rule and model architectures comparable across the other datasets. The learning rule hyperparameters are otherwise the same as in §A.1.

The SimCLR model function code can be found here: `https://github.com/neuroailab/lr-identify/blob/main/tensorflow/Models/simclr_model.py`.

## A.3 Word-Speaker-Noise (WSN)

This is the word recognition task described in Kell et al. [2018], but with an updated dataset from Feather et al. [2019] that consists of 793 word class labels (and a null class when there is no speech), with the inputs being cochleagrams generated from waveforms of speech segments (from the Wall Street Journal [Paul and Baker, 1992] and Spoken Wikipedia Corpora [Köhn et al., 2016]) that are superimposed on AudioSet [Gemmeke et al., 2017] background noises. There are 5,810,600 training cochleagrams and 369,864 test set cochleagrams, where

each cochleagram is of dimensions $203 \times 400 \times 1$. The architectures trained on this task are the same as in §A.1 (with the exception that their final readout layer consists of 794 output units as opposed to 1000 output units), namely, AlexNet, AlexNet-LRN, ResNet-18, ResNet-18v2, ResNet-34, and ResNet-34v2. The learning rule hyperparameters and L2 regularizations are otherwise the same as in §A.1.

The WSN model function code can be found here: `https://github.com/neuroailab/lr-identify/blob/main/tensorflow/Models/audionet_model.py`.

## A.4 CIFAR-10

This dataset [Krizhevsky, 2010] consists of 50,000 training images and 10,000 test set images, all of which are of dimensions $32 \times 32 \times 3$. We applied standard CIFAR-10 data augmentations used by the Keras library of random height and width shift, followed by a random horizontal flip. The architectures trained on this task are KNet4, KNet4-LRN, KNet5, and KNet5-LRN. KNet4-LRN is due to Krizhevsky [2012], and the 5 layer variant KNet5 we use adds an extra convolution layer ($5 \times 5$ kernel size with 128 output channels) with a ReLU followed by average pooling with a kernel size of $3 \times 3$ and a stride of $2 \times 2$. The models use an L2 regularization of $1 \times 10^{-5}$ for all learning rules, except for IA which used an L2 regularization of $1 \times 10^{-12}$ (L2 regularization in IA is an important metaparameter for its learning stability).

The base learning rate for SGDM and IA is set to 0.01 for a batch size of 256, and linearly rescaled for the remaining batch sizes. For FA and Adam across all models, this base learning rate is 0.001. Unlike the other datasets, we found the best top-1 performance by setting the base learning rate to be constant for the entire 100 epochs of training with no learning rate warmup.

The KNet model function code can be found here: `https://github.com/neuroailab/lr-identify/blob/main/tensorflow/Models/knet.py`.

# B  Classifier Training Details

## B.1  Feature Selection Procedure

As there are 1,056 training experiments across all learning rules, each model is trained for 100 epochs on its own Tensor Processing Unit (TPUv2-8 and TPUv3-8), saving checkpoints every 5 epochs, resulting in 21 samples in a single trajectory. Once all the models are trained, we generate on GPU their observable statistics (per layer) from their saved checkpoints, on the respective dataset's validation set (averaged across all validation stimuli). The resultant dataset of generated observables therefore consists of 20,736 total examples. For each example, an observable statistic is *not* used if and only if it had a divergent value during the course of training for any task, architecture, or learning rule hyperparameter, since all network parameters and units (as well as loss function and, when applicable, categorization performance) were well defined across these. Since each example in this resultant dataset is the concatenation of each observable statistic's trajectory, this corresponds to a total of at most 504 feature dimensions.

We additionally include the ternary indicator layer position ("early", "middle", or "deep"), determined by which third of the total number of layers for a given model each of its layers belongs to. For SimCLR, the two contrastive head layers are always assigned to "deep". This mapping for each model can be found in the `group_layers()` function defined here: `https://github.com/neuroailab/lr-identify/blob/main/tensorflow/Models/model_layer_names.py`.

## B.2  Parameter Selection Procedure

We run all classifiers on ten category-balanced train/test splits, which are always (with the exception of architecture, learning curriculum, and task holdouts in Figs. 2(b), S3, S4) 75%/25% in proportion. For each train/test split, we perform five-fold stratified cross-validation for each classifier's parameters. The SVM and Random Forest use `scikit-learn` 0.20.4.

For the SVM, we use the `svm.LinearSVC` function in `scikit-learn`. PCA can be applied to the trajectories of the continuous valued observables as a preprocessing step (namely, excluding the ternary feature of layer position), and the binary decision to use it is chosen at the cross-validation stage. Specifically, when PCA is applied, the number of components $\in \{10, 20, 30, 40, 50, 100, 200, 300, 400, 500, 600\}$. The strength of the regularization parameter $C \in \{1.0, 50, 500, 5 \times 10^3, 5 \times 10^4, 5 \times 10^5, 5 \times 10^6\}$.

For the Random Forest, we use the `ensemble.RandomForestClassifier` function in `scikit-learn`. We allow for the number of trees in each forest to be $\in \{20, 50, 100, 500, 1000\}$. We allow the number of features to consider at each split of an internal node to be $\in \{n_{feats}, \log_2(n_{feats}), \sqrt{n_{feats}}\}$, where $n_{feats}$ is the total number of input features.

For the Conv1D MLP, we use TensorFlow 1.13.1 to train a two-layer neural network that consists of a learned 1-layer 1D convolution, followed by a ReLU and (optional) pooling prior to the fully connected categorization layer. We use the Adam optimizer [Kingma and Ba, 2014] to train the classifier on a separate Titan X GPU per split for 400 epochs. The Adam learning rate is set to be $\in \{1 \times 10^{-3}, 1 \times 10^{-4}\}$. The train batch size is set to be $\in \{512, 1024\}$. The kernel size of the 1D convolution is set to be a fraction of the trajectory length $\in \{3 \times 10^{-3}, 7 \times 10^{-3}, 5 \times 10^{-2}, 0.25, 0.5, 1.0\}$. The strides of the 1D convolution is set to be $\in \{1, 2, 4\}$. The number of output filters of the 1D convolution is set to be $\in \{20, 40\}$. The binary choice of a pooling layer between the 1D convolution and fully connected categorization layer is chosen during cross-validation and if used, can be either 1D max pooling or 1D average pooling. L2 regularization is set to be $\in \{1 \times 10^{-4}, 0\}$.

The classifier code can be found in: `https://github.com/neuroailab/lr-identify/blob/main/fit_pipeline.py`. The classifier cross-validation parameter ranges can be found in: `https://github.com/neuroailab/lr-identify/blob/main/cls_cv_params.py`.

## B.3 Variability of Generalization

The second aim of this study is to understand whether the approach described above generalizes to selected variations in the available data. We introduce two variations: (1) **trajectory subsampling**, in which data is taken for part of the model learning trajectory, and (2) **holdouts**, in which all but one "animals", "curricula", and "tasks" are used only during training and the remaining one is used only during testing. (1) investigates whether our discriminative approach is successful when only part of the model training process is available, and (2) investigates where it transfers successfully to unseen classes of input types. These two variations address the fact that neural and behavioral data is usually *not* collected throughout the developmental trajectory of the animal's entire lifespan as well as generalization (particularly) to new animals and curricula.

### B.3.1 Trajectory Subsampling

Trajectory subsampling is performed to represent data taken at a low frequency, data taken for short periods of time, and data taken at different points in the training process. All of these are defined relative to our "full" model learning trajectory, which is generated from epochs $E = \{0, 5, 10, \ldots, 95, 100\}$ spanning the entire 100 epochs, where $|E| = 21$ samples since we sample every 5 epochs during model training. We note that this is inherently subsampled, but we find that it is a good proxy for the entire model learning trajectory while avoiding the prohibitive computational complexity of generating observable statistics for each step across the 100 epochs of every model training process.

We therefore consider a subsampled dataset to be generated from epochs $E' \subset E$. We define three quantities to represent this limitation. **Subsample start position** is the epoch in the model learning trajectory from which the subsampled dataset starts. **Subsample period** is the space between two consecutive samples of the learning trajectory (in units of model training epochs). We consider three subsample periods of 5 epochs (the original value), 15 epochs, and 25 epochs, which correspond to "Dense Subsampling", "Intermediate Subsampling", and "Sparse Subsampling" in Fig. 4(a), respectively. Finally, **subsample proportion** is the proportion of the number of samples chosen for the trajectory subsample relative to that of the overall model learning trajectory (21 total samples). In Fig. 4(a), we represent this as a percentage of the total trajectory, but the number of samples this corresponds to in the "Dense Subsampling" regime is 2, 4, 6, 9, 11, 13, 16, 18, and 21 samples (the latter is the full trajectory); in the "Intermediate Subsampling" regime it is 2, 4, and 6 samples; and in the "Sparse Subsampling" regime it is 2 and 4 samples. We use a different *single* random seed (corresponding to a potentially different subsample start position) for each example to keep the size of the dataset the same (in terms of number of examples) as when the full model learning trajectory is present.

### B.3.2 Holdouts

We hold out certain parts of the dataset from training and use them for testing to understand the transfer capability of a classifier that can separate certain learning rules when encountering new learning rules. We start by considering two types of holdouts: **animals** and **curricula**. We consider an "animal" to be a certain architecture (one of ten) and a "curriculum" to be a certain combination of batch size and dataset randomization seed (one of six), and hold them out by using the remaining animals or curricula in the classifier training set and the held-out one in the classifier testing set. Specifically, for each of the ten architectures, we train the classifier on the data for the other nine architectures and test on the remaining one. We do the same for the six batch size/dataset randomization seed combinations. We also consider **task** holdouts to get a sense of the differences in learning dynamics across different tasks, holding out all examples from each one of the four tasks. We then measure the test accuracy for each holdout.

## B.4 Realistic Data Measurements: Unit Subsampling and Noise Robustness

Thus far we have assumed that whenever data is collected, it can be collected perfectly: no noise, no loss of neurons. In the final aim of this study, we address how our approach handles more realistic situations under which neuroscience data is collected, we analyze the noise robustness and sample efficiency of the observable statistics. We consider the ResNet-18 architecture trained on supervised ImageNet and self-supervised SimCLR with either the FA or IA learning rules, using a batch size of 256, model seed of None, and dataset randomization seed of None. For noise robustness, we add Gaussian noise ($\sigma = 0, 0.5, 1, 5, 10$) to the (forward) weight/activation/layer-wise activity change matrix of each layer before generating observable statistics. For sample efficiency, we randomly subsample the units from the matrix at rates of $3 \times 10^{-3}\%$, $2.08 \times 10^{-2}\%$, $1.44 \times 10^{-1}\%$, $1\%$, $10\%$, $25\%$, $50\%$, $75\%$, and $100\%$. We consider test accuracy for the Cartesian product of these two axes. For each Cartesian product, we use the *same* random seed for unit subsampling and Gaussian noise. The number of random seeds depends on the unit subsample fraction based on how many are needed until test performance stabilizes when averaged across random seeds and ten category-balanced 75%/25% train/test splits. Thus, smaller unit subsample fractions will require more random seeds (and are computationally more efficient to generate data from) than larger fractions, which will require fewer. Specifically, we use six random seeds for $3 \times 10^{-3}\%$, four random seeds each for $2.08 \times 10^{-2}\%$ and $1.44 \times 10^{-1}\%$, three random seeds for $1\%$, and two random seeds for each of the remaining unit subsample fractions.

## C  Supplementary Figures

Figure S1: **Learning rule is least able to be *linearly* determined from the activations for any pair of learning rules.** For all six pairs of learning rules, we train a separate classifier to distinguish each learning rule pair per observable measure. Mean and s.e.m. are across ten category-balanced 75%/25% train/test splits. Chance performance in these settings is 50% test accuracy.

(a) Task performance as a confounded indicator of learning rule separability.

(b) Averaging across the trajectory can hurt generalization.

Figure S2: **Learning rule separability controls. (a)** We train classifiers to separate IA vs. SGDM when their top-1 validation accuracy is similar on ImageNet, namely across all learning hyperparameters for ResNet-18, ResNet-18v2, ResNet-34, and ResNet-34v2. Mean and s.e.m. are across ten category-balanced 75%/25% train/test splits. Chance performance in this setting is 50% test accuracy. **(b)** We train classifiers to separate all four learning rules after averaging the statistics across the trajectory. Red lines indicate test set performance of each classifier when trained on the entire trajectory for each respective observable measure, as reported in Fig. 2(a). Mean and s.e.m. are across ten category-balanced 75%/25% train/test splits. Chance performance in this setting is 25% test accuracy.

(a) Heldout architecture

(b) Heldout training curriculum

(c) Heldout task

Figure S3: **Assessing robustness of generalization for individual observable measures (Random Forest).** Let the red line indicate the mean and s.e.m. test accuracy from Fig. 2(a) for the Random Forest trained on each observable measure. **(a)** We hold out *all* examples from each of the ten architectures. **(b)** We hold out *all* examples from each of the six combinations of batch size and dataset randomization seed pair. **(c)** We hold out *all* examples from each of the four tasks. Chance performance is 25% test accuracy.

(a) Heldout architecture

(b) Heldout training curriculum

(c) Heldout task

Figure S4: **Assessing robustness of generalization for individual observable measures (SVM).** Let the red line indicate the mean and s.e.m. test accuracy from Fig. 2(a) for the SVM trained on each observable measure. **(a)** We hold out *all* examples from each of the ten architectures. **(b)** We hold out *all* examples from each of the six combinations of batch size and dataset randomization seed pair. **(c)** We hold out *all* examples from each of the four tasks. Chance performance is 25% test accuracy.

Figure S5: **Sparse subsampling *across* the learning trajectory is robust to trajectory undersampling (SVM).** For each observable measure, the test accuracy of the SVM on a random 25% split of the data. Mean and s.e.m. are across the ten category-balanced 75%/25% train/test splits. Each "full" training trajectory contains 21 samples (corresponding to epochs $0, 5, 10, \cdots, 100$). "Subsample Proportion $Y$%" refers to the number of samples chosen for the trajectory subsample relative to that of the full trajectory (21 total samples). Chance performance in these settings is 25% test accuracy.

Figure S6: **Generalization performance can vary when trained solely on a consecutive portion of the learning trajectory, regardless of classifier.** For each observable measure, the test accuracy on a random 25% split of the data. Mean and s.e.m. are across the ten category-balanced 75%/25% train/test splits. Each training sample contains each consecutive seventh of the trajectory, totalling 7 consecutive portions each consisting of 3 samples (top row), or consecutive third of the trajectory, totalling 3 consecutive portions each consisting of 7 samples (bottom row). Chance performance in these settings is 25% test accuracy.

Figure S7: **Training solely on consecutive portions of the learning trajectory is *not* robust to trajectory undersampling (SVM).** We train the SVM on 75% of examples with access to only one consecutive third of the full trajectory (7 consecutive samples each out of 21 total samples), testing on the remaining 25% of examples from one of the other two consecutive thirds. Red lines denote SVM performance when tested on 25% of examples from the *same* portion of the trajectory as in classifier training for each observable measure, reported in the bottom left row of Fig. S6. Mean and s.e.m. in all cases are across ten category-balanced train/test splits. Chance performance in this setting is 25% test accuracy (denoted by the dotted black line).

Figure S8: **Activations are the most robust to measurement noise and unit undersampling (SVM).** For each observable measure, the SVM test accuracy separating FA vs. IA on ResNet-18 across the ImageNet and SimCLR tasks, averaged across random seed and ten category-balanced 75%/25% train/test splits. The $x$-axis from the left to right of each heatmap corresponds to decreasing levels of noise, and the $y$-axis from bottom to top corresponds to increasing levels of sampled units. Top right corner of each heatmap corresponds to the perfect information setting of §4. Chance performance in this setting is 50% test accuracy.