[Reviews · NeurIPS 2020]

Review 1

Summary and Contributions: This manuscript asks whether it is possible to classify the learning rule that has been used to train a network on the basis of statistics of weights or activations of the network. SVMs, random forests, or a MLP is used to classify between SGD with momentum, Adam, feedback alignment, information alignment. The results reveal that there are signatures of these learning rules in both the weights and activations of networks across training, suggesting that measurements of such quantities may be useful for inferring learning rules, perhaps even from biological systems.

Strengths: This is an interesting and creative direction, and it is understudied in the neuroscience literature. The fact that the different algorithms studied can be categorized based on relatively simple metrics is surprising and interesting. While other studies have focused on inferring quantities determining parameterized learning rules, distinguishing between broad classes of learning rule on the basis of recordings has been less studied. Although there is a long way to go to connect this to biology, it is an interesting new direction.

Weaknesses: The main concern is whether the choices of learning rule are appropriate to draw conclusions. As discussed below, there is a long way to go to connect these to brain recordings. However, given that, I would have liked to see a method that, for instance, distinguishes between supervised, unsupervised, or reinforcement learning, as opposed to between different variants of supervised learning (since none of these variants are going to be exactly what a brain is using). The authors do appear to study with supervised and unsupervised contrastive algorithms, but group them together. Are they distinguishable? The fact that weight medians and norms provide the most predictive power suggests that relatively simple features are enough to differentiate between the learning rules presented, but the conclusion from the last paragraph that electrophysiological measurements will be enough to do this for a real brain is a stretch. We don't have much reason to believe that these four learning rules, and these architectures, are representative. It would be helpful if the authors could provide some intuition as to why simple measures like weight medians are most informative. Why does one expect these different learning rules to yield different medians?

Correctness: The simulations appear to be implemented correctly.

Clarity: The paper is clearly written.

Relation to Prior Work: This is a relatively novel direction. However, more time could be spent contrasting learning hyperparameters (e.g. Lim et al.) and learning rules. With a sufficiently rich parameterization, are the two really different?

Reproducibility: Yes

Additional Feedback: After discussion and author response, I continue to feel positively about this paper.


Review 2

Summary and Contributions: ==========Update after rebuttal=========== I've read the author's rebuttal and the other reviews. The rebuttal did a good job of addressing many of the reviewers' concerns. I maintain that the paper is highly important and should be published. The camera-ready version will benefit from taking into account several of the reviewers suggestions. ===================================== The paper addresses the important question of what aspects of learning rules can be inferred from observing only changes in the activity and weights of a neural network during training, without knowledge of the architecture or loss function. In particular, this paper attempts to distinguish four learning rules—stochastic gradient descent with momentum (SGDM), Adam, feedback alignment (FA), and information alignment (IA)—using a number of simple classifiers (linear SVM, Random Forest, Conv-1D MLP) trained on several summary statistics, aggregated over units within each layer. The experiment is repeated over six architectures trained with both supervised and contrastive learning. The authors found that they were indeed able to classify learning rules from trajectories of observables and evaluated the robustness of this classification in the presence of unit subsampling and noise.

Strengths: The results of this paper are highly significant and novel and relevant to the NeurIPS community. This research problem was discussed last year at the NeurIPS workshop on Neuro-AI where panelists agreed that this question should be a top priority for the field if we are to make progress on biologically plausible learning rules and identifying what learning mechanisms are employed in biological brains. It is particularly interesting that while observables calculated from network activations were not the most important in the noiseless models, they were more robust than the other two measures in the presence of noise and unit-subsampling. This is good news for neuroscientists who generally have access to noisy, subsampled activations and not to the weights or layer-wise activity change.

Weaknesses: * The authors did not employ any perturbation or Hebbian-based learning schemes due to their slowness or inability to learn. In their defence, they claim that FA and IA include some similar mechanisms. However, I am confident that many researchers will specifically want to see the ability to discriminate between Hebbian-based and gradient-based learning rules, which is not present in this paper. * The paper is missing a confusion matrix summarizing how well the different learning rules are distinguished from each other. Figure S1 is in this direction. The text from that figure caption "Feedback alignment (FA) is linearly separable from any other learning rule from the activations observables as well as the other observables measures (c-e). However, for pairwise comparisons between the top performing learning rules, the activations observable is not always a good basis for linear separability, whereas the weight and layer-wise activity change observables are (a,b).” is important enough that I think it should be in the main paper. However, I would really like to see a confusion matrix, not just these pairwise bar plots. I think you could find room in the main text, especially if you used numbered in-text citations instead of author-year. * We know that something like the scale of the activations or weights won’t help us to identify learning rules in biological brains. However, it is possible that simple features that are not time varying may contain some information about learning rules in artificial networks. I would like to see the test accuracy when using statistics calculated over the whole trajectory (e.g. mean across time) to verify that the robustness that you observe is not partially a product of something simple like the scale of the weights or activations.

Correctness: The claims and empirical methodology are correct to the best of my knowledge. I am not familiar with the Gini-impurity feature importance measure and so cannot comment on its appropriateness in this setting. Perhaps one of the other reviews can comment.

Clarity: In general, the paper is well written and clear. However, I think there is room for improvement in the description of the features/statistics used. You start by describing the three features (weights, activations, layer-wise activity change), three functions (raw, abs, sq), and the seven summary statistics, which results in "45 observables for each layer". But 3 * 3* 7 is 63 not 45 so I am already confused. And many of those features not actually analyzed? In the later description of 546 feature dimensions and 21 timepoints, I infer that 26 observables are analyzed, which I assume correspond to the 26 colored bars in Figure 3. You explain that you only use the mean statistic for the activity change feature, but this still doesn’t get us to 26 observables. If Figure 3 corresponds to the total set of observables that were analyzed, I recommend you update the description of their definition to make this more clear. Also, the ordering of observables in Fig 3 seems somewhat random. Some organization would be useful here. minor comments: * line 79: replace identifying with distinguishing * line 121: simCLR is an example of self-supervised learning, not unsupervised. To the extent that these distinctions between unsupervised, self-supervised, semi-supervised, etc are useful, I recommend trying to be consistent with how these terms are used in the literature. * Figure 5 and S6: The use of a diverging colormap is confusing because the middle point (white) of the hot-cold scale has no meaning and the red values are not intended to be categorically different form the blue values. I recommend a standard sequential colormap.

Relation to Prior Work: The authors cite previous work in neuroscience which has been successful at inferring hyperparameters of learning rules from neural activity but has failed to distinguish different classes of learning rules, as demonstrated here in artificial neural networks.

Reproducibility: Yes

Additional Feedback: * In order for such inferences about learning rules to be made from biological brains, the task must be novel (so that there is something to learn). As such, the tasks used here, related to forming visual representation of objects in natural images, will likely not be appropriate since the tested animals will likely already have learned to form representations of natural visual scenes. Eventually we will want demonstrations like the ones in this paper but for tasks that can also be used, for example, in human studies to make the same kind of inferences. * One ethical concern that is not discussed is related to the problem of reverse engineering neural networks. For example, there are several papers making progress on identifying weights and architectures just from probing the outputs of a network. Combined with those results, this work on identifying learning rules from network observables (which could potentially be inferred from outputs) suggests the possibility of being able to infer quite a lot about a model (architecture, weights, learning rule) just from the network outputs. This presents obvious privacy concerns which could be exploited, for example, to generate adversarial attacks or even to recover training data. * D. Rolnick, K.P. Körding, Reverse-engineering deep ReLU networks, International Conference on Machine Learning (ICML) 2020. * D. Rolnick, K.P. Körding, Identifying weights and architectures of unknown relu networks, arXiv preprint arXiv:1910.00744


Review 3

Summary and Contributions: The authors carry out a virtual neuroscience experiment. The purpose of the experiment is to infer the learning algorithm which modifies the parameters of a neural network, from the trajectory it induces. A number of different deep artificial neural network models is trained on ImageNet tasks (both classification as well as contrastive unsupervised learning) with four learning algorithms. While learning proceeds, the authors collect aggregate statistics (over neural activity and weights) which summarize the network state. These aggregate quantities are then used as predictors to train simple classifiers, which attempt to discriminate which of the four learning rules trained the model. This ideal observer approach allows understanding which predictors separate the rules. The results are overall positive for the settings considered. The authors draw a number of interesting conclusions regarding the robustness of the different predictors to noise and sampling. UPDATE: Having read the author's response, I maintain my positive assessment of the paper.

Strengths: - Creative and thought-provoking approach which may inspire future other 'virtual experiments' of the kind. - Well-chosen features have discriminative power and are interpretable, useful analogies. - Well-written, accessible paper, that will probably reach experimentalists and theorists alike.

Weaknesses: - Results with an unsupervised clustering algorithm (oblivious to which of the four learning algorithms an example belongs to) would have been a nice plus. - Ideal observer approach not immediately translatable to a 'non-virtual' experiment. - Generalization experiments are appropriate, but could be improved. In particular all models are trained on the same dataset (admittedly, lifting this constraint might not prove easy) and all network models considered are rather similar.

Correctness: I did not find any methodological problems.

Clarity: The paper is very clear and well written.

Relation to Prior Work: The work by Lim et al. is the most related work that I'm aware of, and it is well discussed here. Furthermore, the paper appropriately cites and discusses a large amount of relevant literature.

Reproducibility: Yes

Additional Feedback:


Review 4

Summary and Contributions: This is a very ambitious paper which empirically tests a question that, on first blush, I was sure was not identifiable: discriminating between different learning rules without knowing the models or objective functions involved in the optimization procedure. The results are strong, and demonstrate that their method generalizes well and the authors predict the level of biological noise that could render the method ineffective. --- Update --- This is a deeply interesting paper with big ideas. I was initially hesitant on the empirical support for these ideas, but I think the author's have great arguments in their rebuttal. Perhaps this experiment is impossible, but what I'd like to see in the final version is for the author's to show that they can reliably discriminate models when their performance is *exactly the same* on ImageNet. I.e., find the weights where all models have 30% top-1 error (or whatever is lowest common denominator) and show that the statistics they gather are still sufficient under this stringent test. End of the day though, I think this is a great paper and potentially the start of a really novel line of research in systems and computational neuroscience.

Strengths: These experiments are exciting and potentially reveal (a) a path towards relating engineered learning rules to those implemented in model systems, (b) the experimental conditions when learning rules are most easily identified in vivo. This work has great potential for high impact in systems and computational neuroscience. I especially enjoyed the experiments evaluating the method in different amounts of noise, which yields predictions that can guide systems neuroscience experiments.

Weaknesses: This paper has stuck with me, and I do want to emphasize just how interesting I find it. I am very much in favor of it, but the following list of weaknesses is holding me back from backing its acceptance. Broadly, I need more convincing that (1) discrimination is not trivially due to differences in learning alg performance, (2) how learning algorithm vs. architecture can ever be dissociated in model organisms, and (3) *why* would differences at the level of weights (fig 3) be indicative of different learning algorithms in a way that cannot be deduced via first principles (related to point 2). (4) How the learning dynamics of a model -- initialized from random weights -- relate to the learning dynamics of a model organism, which begin from a epigenetically inherited "initialization". 1. I am suspicious that the ability to discriminate between learning algorithm is driven by differences in their performance on imagenet. While it's not obvious to me *how* this would work, it seems plausible that learning algorithm differences e.g. weight norms/raw medians could be predictive of performance. I think that the authors need to control for this. One way to test whether model performance is a confound in learning algorithm discrimination is to see if they can predict learning rule effectiveness on imagenet according to the same statistics they report in the paper. The key problem here is that if performance is driving discrimination, then I dont see why you would need to collect these statistics for discriminating learning algorithms. Simply train/test on imagenet, maybe collect the pattern of decisions models trained with each learning alg produces, then compare to behavior/neural data. 2. The authors plot the importance values from RF on learning alg discrimination, showing that certain statistics are more informative than others for doing this. It is still not clear to me why this should be the case? Do you have any computational reasoning for why different learning algorithms, like SGD vs. FA, would be discriminable at the level of synapses? How could differences in norm, for instance, discriminate between learning algorithms, when these are surely just as much a function of architecture and hyperparameters (such as learning rate and regularization)? As another example of the difficulty here, neurons have recurrent dynamics, yet the experiments here deal with learning algorithms for static models. If they *were* to adopt recurrent learning algorithms intead, perhaps there would be something related to the types of attractors preferred by different learning algorithms that could be used to discriminate them. But in this case, I am really struggling to imagine how this method could say that primate visual system is using Adam and not SGD+Momentum or FA but not SGD. The architecture is intrinsically tied to the learning algorithm! 3. Relatedly, you use Adam and SGD+Momentum as two of the learning algorithms. But the only difference between the two is in learning rate, which for the former is computed dynamically over training. Discriminating learning rate seems like a different question of discriminating learning algorithms. Learning rates will have big impacts on the stats at individual synapses, by construction, but I don't understand why learning algorithms must do the same. 4. How does discriminability change when initializing from relatively good weights, rather than a random init? (Broken record now) how does discriminability change when changing model regularization or learning rate (not just rescaling, but actually increasing or decreasing these hyperparameters)? Can a model trained with one set of hyperparameters generalize to different learning rates/regularizations? To conclude, this is fascinating work, but I am not sure whether these simulations are really relevant to biology.

Correctness: Yes.

Clarity: This is a very well written paper.

Relation to Prior Work: Yes.

Reproducibility: Yes

Additional Feedback:

[Author Response · NeurIPS 2020]

We thank all reviewers for careful reading & positive comments, including **R1**: "that different algorithms can be categorized based on relatively simple metrics is surprising & interesting"; **R2**: "the results...are highly significant and novel and relevant to the NeurIPS community"; **R3**: "creative and thought-provoking approach which may inspire future other 'virtual experiments' of the kind"; **R4**: "this work has great potential for high impact in systems and computational neuroscience". We now address major reviewer concerns below. ★**How biological are the architectures, task, & learning rules evaluated ...? Why these particular choices? (R1,R2,R3,R4)**: We chose NN architecture types & training datasets that have been shown in comp. neurosci. literature to make good models of neural response patterns in primate electrophysiology & human fMRI data. We test learning rules that have competitive ML performance that cannot be ruled out by performance characteristics alone (e.g. simple hebbian rules). We use supervised & self-supervised learning objectives (without need for Imagenet category labels), & a range of different specific NN architectures, to model the fact that the loss function & architecture best suited to understanding a given brain area are generally partially, but far from exactly, known. Our work's goal is to identify statistics that will allow identification of the learning rule, *invariant* across the variability due to these types of unknowns. **R3**, good point about varying datasets / architecture classes. We've obtained results for shallow architectures with CIFAR-10 dataset – biologically, perhaps interpretable as expanding project scope to simpler *non-primate* (e.g. mouse) visual systems. We also have results for networks trained on *auditory* stimuli, using the AudioSet dataset – showing our approach holds across multiple sensory modalities with the *same* classifier. Will include these results in revision. We hope in the future to also broaden scope to e.g. RL models, as suggested by **R1**. ★**"... suspicious that [discrimination power] is driven by differences in Imagenet performance..." (R4)**: Important question. As shown in Fig S1, all learning rules except feedback alignment (FA) have high overlap in performance across hyperparameters; performance differences due to architecture swamp those from learning rule, e.g. FA aside, Alexnet with best learning rule performs $\ll$ Resnet-34 with worst learning rule. Thus, performance is a highly confounded indicator of learning rule, a key point we should have emphasized, so will move to main text as **R2** suggests. Also, we want to address experimental situations where performance is not directly measurable (animal behavior is often harder than e-phys!); & to allow for the possibility of unsupervised learning objectives not optimized for specific performance goals. Thus, it is important & nontrivial to identify features that are robust across architecture & objective fns, & have direct physical analogues in experimental measurement. ★**"The authors [show] that certain statistics are more informative than others ... not clear why this should be the case?" (R4)**: The primary intuition that certain aggregate statistics could be useful for separating learning rules comes from studying the learning dynamics of single layer perceptrons, where activation mean is a typical choice [eg. Werfel et al. 2004]. But in deep NNs, no theory yet allows us to derive optimal statistics mathematically, motivating our empirical approach. We thus included a variety of potential observables that might more robustly characterize non-linear network effects, & thus enable the classifier to *discount* differences when needed. Ideally in the future we can combine better theory with our method to sharpen feature design. We will improve discussion of this in revision. ★**"...neurons have recurrent dynamics, experiments here [only use feedforward] models... architecture is intrinsically tied to the learning algorithm!" (R4)**: We have tested our approach on recurrent convolutional models [Nayebi et al. 2018, Schrimpf et al. 2019] – just not at such large scale as the included results, since such networks are very resource-intensive. However, outcomes don't change conclusions at all, will include what we have in revision. Importantly: a main takeaway of our paper is that architecture is in some sense *not* necessarily intrinsically tied to the learning rule; otherwise, we would not have been able to reliably separate learning rules across the range of architectures considered. ★**"Relatedly, you use Adam & SGD+Momentum ...Discriminating learning rate seems like a different question of discriminating learning algorithms." (R4)**: First-order learning rules are basically characterized by two choices, namely how parameter updates are made as a function of (1) (high-dimensional) direction of gradient tensor, & (2) the magnitude of gradient tensor. Item (2) is directly tied to learning rate policy, & as adaptive methods can yield significant (if hard to predict) differences in trainability across various architectures & datasets, learning rate policy is an integral part of the learning rule. Our choice of candidate rules tested the ability of our approach to handle variation of *both* aspects. ★**"Does discriminability change when initializing from relatively good weights, rather than random?" (R4)**: While we're not exactly sure how to initialize from good weights in a task agnostic way (we used standard best practices for init), we *did* examine training the classifier solely on different portions of training trajectory, including only using late-time checkpoints after network performance stabilized – this somewhat approximates idea of using "good" weights. We found largely consistent results (Fig. S4). Interesting question for follow up work! ★**"Can a model trained with one set of hyperparameters generalize...?" (R4)**: In all reported results, we widely varied not only architecture & loss function, but also learning hyperparameters such as learning rates/regularizations (see supplement for details). We then considered two types of classifier accuracy evaluations. First, we performed standard cross-validation, e.g. random non-overlapping train/test splits. High accuracy here shows classifiers work across new mixed combinations of architecture, objective function, & learning hyperparameters. We also performed tests that held out entire classes of input types, to explore strong generalization. For example, we did architecture hold-outs, training on some architectures then testing on others, & our method still performed well in this crucial case (Fig 2b). Also see Figs. S2-3 for other such generalization tests. ★**Other comments (R1-R4)**: We cannot address all remaining comments due to space limitations, but will address them in revision, especially **R2**'s stylistic suggestions.

[Meta-Review · NeurIPS 2020]

All four referees support accept. In general, reviewers agree that this submission is extremely novel and highly relevant to both neuroscience and ML researchers alike. Author response convincingly addressed concerns several reviewers raised about experiments. Thus, I strongly recommend accept, but encourage authors to incorporate reviewer feedback in camera ready.